# Unusual Presentation of *SET::NUP214*-Associated Concomitant Hematological Neoplasm in a Child—Diagnostic and Treatment Struggle

**DOI:** 10.3390/ijms241914451

**Published:** 2023-09-22

**Authors:** Yaroslav Menchits, Tatiana Salimova, Alexander Komkov, Dmitry Abramov, Tatiana Konyukhova, Ruslan Abasov, Elena Raykina, Albert Itov, Marina Gaskova, Aleksandra Borkovskaia, Anna Kazakova, Olga Soldatkina, Svetlana Kashpor, Alexandra Semchenkova, Alexander Popov, Galina Novichkova, Yulia Olshanskaya, Alexey Maschan, Elena Zerkalenkova

**Affiliations:** 1Dmitry Rogachev National Medical Research Center of Pediatric Hematology, Oncology and Immunology, Samora Maschela Str., 1, 117998 Moscow, Russiatatiana.konyuhova@fccho-moscow.ru (T.K.); ruslan.abasov@fccho-moscow.ru (R.A.);; 2Abu Dhabi Stem Cells Center, Mahdar Qutouf Str., 25, Abu Dhabi 22404, United Arab Emirates; alexander.k@adscc.ae

**Keywords:** pediatric leukemia and lymphoma, *NUP214*, fusion genes, HSCT

## Abstract

Simultaneous multilineage hematologic malignancies are uncommon and associated with poorer prognosis than single-lineage leukemia or lymphoma. Here, we describe a concomitant malignant neoplasm in a 4-year-old boy. The child presented with massive lymphoproliferative syndrome, nasal breathing difficulties, and snoring. Morphological, immunocytochemical, and flow cytometry diagnostics showed coexistence of acute myeloid leukemia (AML) and peripheral T-cell lymphoma (PTCL). Molecular examination revealed a rare t(9;9)(q34;q34)/*SET::NUP214* translocation as well as common TCR clonal rearrangements in both the bone marrow and lymph nodes. The disease showed primary refractoriness to both lymphoid and myeloid high-dose chemotherapy as well as combined targeted therapy (trametinib + ruxolitinib). Hence, HSCT was performed, and the patient has since been in complete remission for over a year. This observation highlights the importance of molecular techniques for determining the united nature of complex *SET::NUP214*-positive malignant neoplasms arising from precursor cells with high lineage plasticity.

## 1. Introduction

Acute myeloid leukemia (AML) is a heterogeneous group of neoplasms originating from transformed hematopoietic stem cells (HSCs) or more committed myeloid progenitors. It is classified according to the disease’s molecular genetics and the lineages involved in oncogenesis. In children, AML accounts for approximately 20% of acute leukemia cases [1]. In general, children with AML have worse prognoses than those with acute lymphoblastic leukemia, though overall survival has improved significantly over the last 30 years [2].

Peripheral T-cell lymphoma (PTCL) is a heterogeneous group of non-Hodgkin’s lymphomas (NHLs). PTCL accounts for 5% to 15% of NHL cases [3,4]. It develops from post-thymic T and natural killer (NK) cells due to dysregulation of *TCR*-related, costimulatory, and cytokine signaling pathways [5,6]. The 5th edition of the World Health Organization Classification of Hematolymphoid Tumors includes nine groups of mature T- and NK-cell neoplasms based on their cell of origin/differentiation stage, clinical scenario, primary disease localization, and cytomorphology, among which anaplastic large cell lymphoma (ALCL), nodal lymphoma of follicular-helper T-cell (TFH) origin, and peripheral T-cell lymphoma, not otherwise specified (PTCL-NOS) are the three most common subtypes [7]. Mature T-/NK-cell lymphoid proliferations and lymphomas have aggressive clinical courses and inferior outcomes [4].

Thus, AML and PTCL are diverse malignant neoplasms that arise from two distinct lineages and have distinct cytogenetic properties, cellular phenotypes, and leukemogenic events. They can occur in one patient but usually in the scenario of therapy-related acute leukemia in NHL survivors [8,9,10]. In contrast, the simultaneous development of AML and lymphoma in patients without prior exposure to chemotherapy, radiotherapy, or hematopoietic stem cell transplantation is a rare event. Single cases have been described for NHL occurring concurrently with either AML, chronic myeloid leukemia, or *PDGFRB*-positive hypereosinophilia; despite their different morphological and immunophenotypic natures, the common origin of cells in the bone marrow (BM) and lymph node was shown, but only in approximately half of cases [11,12,13,14]. The molecular background of such events is poorly understood but is of great fundamental interest. In the current study, we describe a case of *SET::NUP214*-positive malignant neoplasm in a pediatric patient presenting in a form of simultaneous occurrence of AML and PTCL and verify the common nature of cells in the BM and lymph node using several molecular techniques.

## 2. Results

### 2.1. Case Description

A 4-year-old boy was admitted to the hospital with enlargement of neck lymph nodes, snoring, nose breathing difficulty, sweating, weight loss, and frequent respiratory infections. Physical examination revealed massive lymphoproliferative syndrome with hypertrophy of adenoid tissue and tonsils, generalized lymphadenopathy, and hepatosplenomegaly.

Blood testing showed a white blood cell (WBC) count of 22.9 × 10^9^/L, platelet count of 131 × 10^9^/L, and hemoglobin level of 10.2 g/L. A blood differential analysis revealed 15% blasts, 18% segmented neutrophils, 46% lymphocytes, and 21% monocytes. Lactate dehydrogenase was increased to 551 U/l.

#### 2.1.1. BM Aspirate Examination

A myelogram revealed 23% anaplastic blast cells (Figure 1a) positive for myeloperoxidase (MPO) (Figure 1b) and Sudan Black (Figure 1c), with a weakly diffuse periodic acid–Schiff (PAS) reaction in some blast cells (Figure 1d). Nonspecific esterase was negative in most blast cells (Figure 1e). This pattern was consistent with acute myelomonocytic leukemia. The cerebrospinal fluid was intact.

However, BM immunophenotyping revealed a relatively complicated cell distribution. In addition to mature neutrophils (CD66b^+^), B-lymphocytes (CD19^+^CD45^bright^), T-lymphocytes (CD3^+^CD45^high^TCRαβ^+^), and NK cells (CD56^+^CD45^bright^), there were several cell populations that warranted focused attention (Figure 2). Early myeloid blasts (2.5% of nucleated cells (NCs)) with CD45^dim^ expression were also characterized by bright CD7, heterogeneous CD33, CD34, and CD117 positivity, and expression of CD2, CD5, CD99, HLA-DR, and CD15. Based on the lymphoid coexpression and general immunophenotypic patterns, the cells were recognized as definitely leukemic. In addition, granulocytic maturation was found to arise from these blasts. At the same time, lymphoid and monocytic compartments also did not appear to be definitely normal although they lacked clear immunophenotypic aberrations (Figure 2). Thirty-four percent of NCs belonged to different stages of monocytic differentiation and were clearly ascribable to immature and mature fractions, as deemed by CD14 expression (Figure 2). In addition to the high proportion of monocytic cells in the BM, partial CD7^dim^ expression in the more immature part of the population was considered a feature of the leukemic origin of this compartment. Additionally, two populations of γδ-T cells were detected (Figure 2). The first population demonstrated high expression of surface CD3, CD5, CD7, and CD2. As expected, these cells were CD4/CD8-negative and displayed high expression of γδ-type TCR. These lymphocytes comprised 11% of NCs and were considered normal. The second population (3.5% of NCs) expressed all T-lineage markers (CD7, CD2, CD5, CD3, TCRγδ) at lower levels than “normal” γδ-T lymphocytes. These cells also had CD45^dim^ and CD99^high^ expression; they were CD4/8-negative and CD48-negative. Although the two described populations of γδ-T cells did not have distinct borders on dot plots (Figure 2), they were considered different (first—normal, second—leukemic) and were isolated separately using flow cell sorting. Simultaneously, immature and mature monocytes were sorted for further molecular investigation.

#### 2.1.2. BM Trephine Biopsy

A BM trephine biopsy showed that intertrabecular spaces were uniformly filled with hypercellular hematopoietic tissue. The cellular component was represented mainly by immature elements of granulopoiesis, a large number of promyelocytes, and myeloblasts located predominantly in the centers of intertrabecular spaces, as determined by MPO expression. The erythropoietic lineage was represented by single spectrin- and glycophorin-positive cells not forming erythron islets. Megakaryocytes were represented by small cells and megakaryoblasts that expressed CD61. Lymphoid clusters positive for CD3, weakly and irregularly positive for CD5 and CD2, and negative for CD20/Pax5, TdT, and CD30 were present. A small number of CD8-positive cells were also found.

#### 2.1.3. Lymph Node Examination

The patient underwent biopsy of a cervical lymph node. The morphological data showed a complete loss of topographic structure of the lymphoid tissue of the lymph node due to a monomorphic population of moderately atypical medium-sized lymphocytes infiltration (Figure 3a,b). Mitotic activity was low. On immunohistochemical examination, the infiltrating cells were represented by two populations—CD3^+^/CD7^+^/CD5^+^ lymphocytes, with weak and uneven CD2 expression, negative for CD8, CD4, PAX5, TdT, CD68, CD14, CD56, and CD163, consistent with the diagnosis of peripheral T-cell lymphoma (Figure 3d), and MPO-positive cells (Figure 3c).

Considering the morphological, cytochemical, and immunological data, the coexistence of two neoplastic populations was suggested, as represented by a peripheral T-cell lymphoma, not otherwise specified, and an acute myelomonocytic leukemia. The diagnostic examinations are summarized in Table 1.

### 2.2. Cytogenetic and Molecular Findings

The BM aspirate was subjected to cytogenetic and FISH analyses. G-banded chromosome analysis revealed an abnormal male karyotype 46,XY,t(3;11)(q21;q23)/46,XY (Figure 4). FISH analysis was negative for *KMT2A*, *PDGFRB*, *ETV6*, *FGFR1*, *JAK2*, *IGH*, *BCL6,* and *PDGFRA* rearrangements. T(9;22)(q34;q11)/*BCR::ABL1* and *ATM* loss were not found. The FISH results are summarized in Table A1.

Sanger sequencing of the BM aspirate revealed a pathogenic *CBL* mutation (c.1186T>C NM_005188.3, p.Cys396Arg NP_005179.2). The mutation was also found in the lymph node biopsy material but was absent in DNA isolated from the nails, which confirmed its somatic status (Figure 5).

The targeted next-generation sequencing (NGS) approach did not reveal any mutations in genes associated with immunodeficiencies. However, deletions at 3q13.11 and 11q14.2 were found (210.412 kb and 2.091 Mb, respectively). Targeted NGS also detected a 9q34.12-13 deletion (2.270 Mb).

Based on whole-transcriptome analysis, no fusion transcripts corresponding to the t(3;11)(q21;q23) translocation were found. However, the *SET::NUP214* chimeric transcript was detected (Figure 6a), which is in line with the 9q34 deletion found using DNA-based NGS. The *SET::NUP214* fusion transcript was confirmed via direct Sanger sequencing of the BM sample (Figure 6b). The *SET::NUP214* fusion gene was also amplified using direct genomic DNA PCR (for primers, see Table A2) and validated using amplicon NGS (Table A3). A patient-specific primer pair was designed for the DNA breakpoint junction region (Table A2) and used for detection using the lymph node biopsy material and isolated BM population DNA. The *SET::NUP214* fusion gene was detected in both the lymph node and BM aspirate, as well as in all BM flow-sorted cell populations, but it was absent in the nail DNA (Figure A1).

For further confirmation, *NUP214* status was assessed using break-apart FISH which revealed 5′-partial deletion in 90% of BM nuclei (Figure 6c), confirming *SET::NUP214* fusion formation due to del(9)(q34.11q34.13) interstitial deletion.

### 2.3. TCR/BCR Repertoire

The spectrum of *TCR*/*BCR* loci rearrangements was assessed in whole BM, lymph node biopsy samples, and isolated BM populations. Major detected clonal rearrangements (VAF > 5% in the BM) included three *TRB* DJ-junctions, two *TRD* VDJ-junctions, and one nonfunctional *IGH* DJ-junction (Table A3). According to the distribution of *TRB* clonal rearrangements in single cells, one of the alleles of the *TRB* locus simultaneously exhibited two rearrangements, which was rather nontrivial (Figure A2). All rearrangements were present in each analyzed sample, indicating their common origin. However, some variations were observed in fractions of the detected rearrangements between the samples (Figure 7).

Considering the molecular findings, the final World Health Organization (WHO) diagnosis of a peripheral T-cell lymphoma, not otherwise specified, and an acute myeloid leukemia with other defined genetic alterations was made.

### 2.4. Treatment and Outcome

Considering the massive lymphoproliferative syndrome and high risk of upper respiratory tract obstruction, cytoreduction with prednisolone 30 mg/m^2^/day (1 day) and 60 mg/m^2^/day (2–4 days) was initiated, followed by CHOP chemotherapy: prednisolone 100 mg/m^2^/day (1–5 days), vincristine 1.5 mg/m^2^ (1 day), cyclophosphamide 750 mg/m^2^ (1 day), and doxorubicin 50 mg/m^2^ (1 day). Improved speech and nasal breathing were noted as well as decreased hepatosplenomegaly and lymphadenopathy.

On Day 12 after CHOP, rapid inguinal lymph node enlargement and hepatomegaly were again noted, and lactate dehydrogenase levels increased. Therefore, high-dose chemotherapy potentially active against both lymphoid and myelomonocytic lineages was initiated: dexamethasone 20 mg/m^2^/day (1–4 days), cytarabine 1000 mg/m^2^ twice a day (1–2 days), cisplatin 100 mg/m^2^/day (1 day), and etoposide 150 mg/m^2^/day (1–2 days). The patient tolerated the chemotherapy well. After the block, a decrease in leukocytosis and monocytosis, as well as a decrease in the number of blast cells, was observed in the peripheral blood. There were no control BM punctures, as lymphadenopathy and hepatomegaly persisted.

The patient was scheduled for hematopoietic stem cell transplantation (HSCT) as the only possible treatment option for the underlying disease. Given the ineffectiveness of standard high-dose chemotherapy and *CBL* mutation presence, combined targeted therapy was started on Day 46 after diagnosis (trametinib 0.25 mg/day + ruxolitinib 15 mg/day) in an attempt to target the hyperactivated pathways normally inhibited by *CBL* and to reduce the tumor mass before HSCT. However, no response was achieved, and on Day 13, peripheral lymph node enlargement and hepatosplenomegaly worsened, hyperleukocytosis with blast cells appeared, and lactate dehydrogenase increased.

Thus, salvage therapy was initiated with fludarabine 30 mg/m^2^/day (1–5 days), high dose cytarabine 2000 mg/m^2^/day (1–5 days), and idarubicin 10 mg/m^2^/day (1–2 days). This chemotherapy was well tolerated, and a significant reduction in the size of the tonsils and lymphadenopathy (clinically and radiologically) was achieved. Blasts disappeared from the peripheral blood, but hepatosplenomegaly persisted.

On Day 74 after diagnosis, the patient underwent allogeneic HSCT with *TCR* αβ/CD19 depletion from a haploidentical mother after conditioning, comprising total marrow and lymphoid irradiation and melphalan. The post-transplant period was uneventful, engraftment occurred on day +14, and complete remission was achieved. On Day 180 after HSCT, no lymphadenopathy or organomegaly persisted according to CT scans (Figure 8). The patient has remained in CR for 16 months after HSCT.

## 3. Discussion

As a general rule, AML and lymphoma arise from different malignant progenitors and rarely occur simultaneously. There have been many reports of therapy-related AML after previous exposure to cytotoxic drugs or radiotherapy used in the treatment of other malignancies or autoimmune disorders; the incidence of therapy-related AML in patients with breast cancer, non-Hodgkin’s lymphomas, and Hodgkin’s lymphoma is noticeable [15,16,17]. In addition to previous toxic therapeutic or environmental exposure, specific inherited conditions such as Li-Fraumeni and BRCA syndromes may result in the co-occurrence of several tumors in one patient in the form of multiple primary tumors. However, acute leukemias are uncommon in multiple-primary-tumor patients [6], and concurrent hematologic malignancies are extremely rare [11]. Here, we present a case of simultaneous PTCL and acute myelomonocytic leukemia.

The patient presented with lymphoproliferative syndrome, hyperleukocytosis with monocytosis and peripheral blastosis, and increased lactate dehydrogenase levels. Autoimmune lymphoproliferative syndrome (ALPS), transformed juvenile myelomonocytic leukemia (JMML), AML, and peripheral T-cell lymphoma were considered in the differential diagnosis.

As lymphoproliferation was the leading symptom, the primary diagnostic search began with ALPS. ALPS is a disease characterized by immune dysregulation caused by an inability to regulate lymphocyte homeostasis via abnormalities in lymphocyte apoptosis or programmed cell death. This deficiency causes a lymphoproliferative disease with clinical manifestations such as lymphadenopathy, hepatomegaly, splenomegaly, an increased risk of lymphoma, and autoimmune diseases (typically involving hematopoietic cell lines). ALPS is characterized by an increase in TCRα/β^+^CD4^−^CD8^−^ cells in the peripheral blood and tissues [18]. Despite the clinical presentation of lymphadenopathy and hepatosplenomegaly, double-negative T lymphocytes were at normal levels, and the diagnosis of ALPS was excluded.

JMML is a rare aggressive childhood myeloid tumor affecting 1.2 people per million [19,20] and exhibiting various clinical signs. It typically affects infants and young children and presents with fever, splenomegaly, a high WBC count, and peripheral monocytosis, though these nonspecific symptoms can all be due to bacterial or viral infections [21]. Another typical JMML sign is an increase in fetal hemoglobin synthesis. Deregulation of the intracellular Ras signal transduction pathway, which is induced in >90% of patients by mutations in one (or, rarely, more than one) of five primordial genes (*PTPN11*, *NRAS*, *KRAS*, *NF1*, or *CBL*), is the shared molecular characteristic of JMML [22]. Our patient carried a somatic *CBL* mutation (NM_005188.3 c.1186T>C) described once previously as germline pathogenic in JMML and Noonan-like syndrome [23]. However, only germline *CBL* mutations are listed in the WHO 2016 JMML criteria [24], and the vast majority of described somatic *CBL* mutations in JMML are homozygous due to acquired uniparental disomy [25], which was excluded in our case. In addition, no fetal hemoglobin increase was detected. Therefore, the diagnosis of JMML was excluded as well. Although T-cell neoplasms have been described in patients with JMML, a concomitant presentation would be extremely unusual [26].

The diagnosis of AML was supported by the morphological and cytochemical assessment of BM smears [27,28,29]. MPO was positive in 17% and Sudan black staining in 27% of blast cells. This morphological and cytochemical pattern was consistent with the diagnosis of acute myelomonocytic leukemia. However, the results of BM immunophenotyping did not correlate with those of cytomorphology. Although myeloid blasts were detected using flow cytometry, several other tumor populations were found. Two subpopulations of mature γδ-T lymphocytes with different immunophenotypes were present, one of which was considered leukemic by its antigen expression profile. However, the same molecular aberrations were demonstrated in both sorted γδ-T subpopulations, confirming that they belonged to one malignant clone. Similarly, morphologically normal granulocytes and monocytes clearly originated from leukemic myeloid progenitors, and according to clonality studies, they shared the same progenitor with γδ-T cells. Although myeloid blasts had several immunophenotypic aberrations, maturing and mature myeloid cells had very slight immunophenotypic deviations. Nevertheless, a direct link between myeloid and T-lineage compartments in the BM was clearly visible (Figure 9).

The RNA sequencing data revealed a *SET::NUP214* rearrangement, which occurred due to 9q34 deletion, and that was also visible in the targeted DNA NGS data. The *NUP214* gene is located in the 9q34.13 chromosomal region. It contains 37 exons and spans 109,078 bp. Its longest transcript is 7568 bp long and encodes a protein consisting of 2090 amino acids. NUP214 belongs to a family of nucleoporins, proteins that form a nuclear pore complex (NPC) for the selective transfer of molecules between the nucleus and the cytoplasm [30]. NUP214 consists of an N-terminal β-propeller domain, a central domain containing two helical motifs responsible for interaction with other nucleoporins and anchoring of the protein in the NPC, and a C-terminal domain containing multiple nuclear transport receptor binding sites [31,32]. NUP214 shows high affinity for the nuclear export receptor CRM1, thus mediating the export of various cargo through the NPC. Wild-type NUP214 deficiency leads to a significant decrease in the efficiency of nuclear proteins and mRNA export [33,34].

Several chromosomal rearrangements involving the *NUP214* gene have been found in hematological malignancies; the most studied is t(6;9)(p23;q34)/*DEK::NUP214* in AML [35], followed by *NUP214::ABL1* in T-ALL [36,37,38,39,40], BCP-ALL [41,42,43], and AML [44], and t(9;9)(q34.11;q32.13)/del(9)(q34.11q32.13)/*SET::NUP214* in acute undifferentiated leukemia [45,46], T-ALL [47,48,49], AML [48,50], MPAL [51], and CML [52]. The latter rearrangement was also extensively studied in vitro in various hematopoietic and leukemia-derived cell lines [48,53].

*SET::NUP214* is a rarer *NUP214* gene rearrangement [37]. The SET gene encodes a multifunctional protein involved in apoptosis, transcription, nucleosome assembly, and histone binding. In eukaryotic cells, SET is expressed in two isoforms, TAF1-α and TAF1-β, which are generated by alternative splicing. Structurally, SET/TAF1-β consists of an N-terminal dimerization domain, a central “headphone”-like domain, and a negatively charged C-terminal acidic domain [54,55]. Both SET isoforms, together with the acid nuclear phosphoprotein pp32, constitute an acetyltransferase inhibitor complex (INHAT) that inhibits p300/CBP-mediated histone acetylation via a mechanism called histone masking. The INHAT SET requires both a “headphone”-like and an acid domain for activity [56]. In addition, SET/TAF1-β can regulate the acetylation of nonhistone proteins independently of the INHAT complex, binding directly to transcription factors such as p53 [57], FOXO1 [58], and the ligand-activated transcription factor glucocorticoid receptor (GR) and negatively regulating their activity [59]. Furthermore, SET suppresses p53-mediated transactivation by inhibiting p300/CBP-dependent H3K18 and H3K27 acetylation, thus blocking both p53-mediated cell-cycle arrest and apoptosis in response to cellular stress [60].

*SET::NUP214* retains most of the *SET* coding sequence fused to the *NUP214* 3′-portion. In our patient, SET exon 7 was fused to *NUP214* exon 18, which is in line with most previously described cases [47,49,50]. Thus, SET-NUP214 exhibits transforming activity via the disruption of both SET and NUP214 functions. First, SET-NUP214 does not interact with GR. Therefore, GR transcriptional activity is not initiated upon glucocorticoid signaling in *SET::NUP214*-positive cells [59], which contributes to resistance to glucocorticoid treatment [59,61,62]. Second, SET-NUP214 interferes with chromatin relaxation by inhibiting p300 histone acetyltransferases (HATs), CREB-binding protein (p300/CBP), and p300/CBP-related factor (PCAF; also known as acetyltransferase 2B/KAT2B). The inhibitory activity of SET on HATs is mediated by its acidic domains, which are retained in SET-NUP214. Abnormal histone hypoacetylation can lead to silencing of genes important for hematopoietic differentiation [63,64]. Third, SET-NUP214 disorganizes nuclear export due to the presence of a large NUP214 portion similar to DEK-NUP214 [65]. Finally, SET-NUP214 and DEK-NUP214 contribute to the upregulation of *HOX* genes, namely *HOXA6-10* and the entire *HOXB* cluster, except for *HOXB7* and *HOXB13*. It is possible that SET-NUP214 recruits DOT1L to the promoter of *HOX* genes, which mediates mono-, di-, and trimethylation of histone 3 lysine 79 (H3K79) [47]. *SET::NUP214* knockdown inhibits elevated *HOXA* expression and reduces cell proliferation in a T-ALL cell line [47].

Overall, the data reported to date suggest that *SET::NUP214* is one of the earliest leukemogenic events [46,47]. As additional evidence for the early origin and high lineage plasticity of *SET::NUP214*-positive malignant cells, we confirmed the common nature of the cells in both lymph nodes and BM using three ways: FISH, PCR, and *TCR* clonality. To the best of our knowledge, this is the first case report of such a rare fusion gene with detailed evidence of the common nature of concomitant hematologic neoplasm [12]. Notably, 90% of the nuclei in the BM carried the 9q34 deletion, as shown by FISH (Figure 6c), though this high percentage of definitely tumorous cells was not revealed using other methods. Direct genomic DNA PCR also identified *SET::NUP214* as a common chromosomal rearrangement in both AML and PTCL. This result suggests that, despite variable morphological, cytochemical, and immunological features, the tumor was indeed an integral cell mass sharing a common fusion gene, *CBL* pathogenic mutation, and common *TCR* clonal rearrangements. Differences in *TCR* fractions may indicate further clonal evolution within the tumor (Figure 7).

## 4. Materials and Methods

Diagnostic procedures included BM morphology and cytochemistry as well as lymph node biopsy and trephine biopsy.

### 4.1. Immunophenotyping

Analysis of the immunophenotype of BM-derived cells was performed using 10-color combinations of monoclonal antibodies (mAbs) according to the diagnostic standards of the Moscow–Berlin group, as described previously [66]. All samples were processed according to the manufacturer’s recommendations. At least 30,000 nucleated cells were collected for each sample using a 3-laser Navios (Beckman Coulter [BC], Indianapolis, IN, USA) flow cytometer. The EuroFlow guidelines for machine performance monitoring were used [67]. Flow-Check Pro Fluorospheres (BC) were used for daily cytometer optimization. The results were analyzed using Kaluza Analysis 2.1 software (BC).

### 4.2. Cell Sorting

Diagnostic samples were processed for cell sorting. Target populations were purified using a BD FACS Aria III flow sorter (Becton Dickinson [BD], San Jose, CA, USA) as described previously [68]. A nonfixative lysis agent (PharmLyse, BD) was used, and the cells were diluted in phosphate-buffered saline (Cell Wash, BD). The cells were sorted in ‘Purity’ mode and collected in Eppendorf tubes containing relevant buffer. A total of 3 to 5 million cells were sorted in duplicate for clonality using NGS.

### 4.3. Cytogenetics and Molecular Genetics

Cytogenetics included conventional GTG-banded karyotyping [69] and fluorescence *in situ* hybridization (FISH; Table A1). The karyotype and FISH results are described according to ISCN2020 nomenclature [70].

Total DNA and RNA from BM aspirates were extracted using an InnuPrep DNA/RNA Mini Kit (Analytik Jena AG, Jena, Germany). DNA was subjected to Sanger sequencing for *KRAS*, *NRAS*, *PTPN11*, and *CBL* mutations [22] and subsequently to targeted NGS. The panel used contains 514 genes associated with primary immunodeficiencies (Appendix A). A DNA library was prepared using a hybridization-based target enrichment method using a custom probe panel (Roche, Indianapolis, IN, USA) according to the manufacturer’s protocol and sequenced using an Illumina NextSeq (Illumina, San Diego, CA, USA). The average depth of target region coverage was 149 reads per bp, and 99% of the bases had a target coverage of at least 30×. The sequence reads were mapped to the human genome reference sequence (GRCh38) [71] and processed using a proprietary bioinformatics data analysis pipeline consistent with international standards [72]. The pathogenicity and clinical significance of the identified variants were assessed in accordance with the criteria of the American College of Medical Genetics and Genomics and the Association for Molecular Pathology [73].

RNA was subjected to whole-transcriptome sequencing (Nextera UltraII Directional RNA kit, NEB, Ipswich, MA, USA) using the Illumina NextSeq (Illumina) platform. The GRCh38 version of the human genome preformatted for alignment pipelines, including the hard-masked PAR regions on chromosome Y, was downloaded from NCBI [71]. Sequenced reads were aligned to the human genome using STAR (ver. 2.10.7b) [74]. Fusion transcripts were detected using Arriba (ver. 2.4.0) [75] and confirmed using Sanger sequencing. The fusion gene breakpoint region was amplified using Q5 DNA polymerase (NEB) and Illumina MiSeq (Illumina) sequenced after NEBNext Ultra II (NEB) library preparation. The direct genomic DNA PCR and RT–PCR primers are listed in Table A2.

### 4.4. TCR/BCR Repertoire

The repertoire of clonal *TCR*/*BCR* rearrangements was assessed as previously described [76]. Briefly, target enrichment was carried out in the first PCR step using a multiplex primer set for the V, D, and J segments of all rearranged *TCR* and *BCR* genes: *TRA*, *TRB*, *TRG*, *TRD*, *IGH*, *IGK*, and *IGL* (MiLaboratory LLC., Sunnyvale, CA, USA, https://milaboratories.com/kits, accessed on 10 February 2022). Each PCR (25 ul) contained a MiLaboratory primer mix, 100 ng of genomic DNA, six units of HS Taq polymerase in 1× Turbo buffer, and dNTPs (0.125 mol/L each) (Evrogen, Moscow, Russia). The amplification profile was as follows: 94 °C for 3 min (initial denaturation), 94 °C for 20 s, 56 °C for 90 s, and 72 °C for 40 s for 10 cycles, followed by 94 °C for 20 s and 72 °C for 90 s for 15 cycles, all at a ramp rate of 0.5 °C/s. The amplicons were cleaned using one volume of magnetic AmPure XP beads (Beckman Coulter), as directed by the manufacturer. Next, Illumina UDI primers were used in PCR to attach sample indices and adapters to the libraries. The final libraries were cleaned as previously described, pooled, and sequenced using a MiSeq (Illumina, USA) instrument (paired-end 150 nt reads). *TCR* and *BCR* rearrangement repertoires were extracted from the sequencing data and converted to the VDJtools format [77] using modified MiXCR software (ver. 4.1.2) [78]. The VDJtools package’s Correct function was used to correct amplification errors. iROAR software [79] was used to correct multiplex-specific quantitative biases. A frequency of 5% was used as the cutoff for identifying leukemic clone-specific rearrangements.

## 5. Conclusions

The combined data of the differential diagnostic search, cytogenetics, molecular genetics, morphology, clinical presentation, and laboratory tests in this study pointed to a concomitant myeloid and lymphoid malignant neoplasm (acute myeloid leukemia, M4 variant + peripheral T-cell lymphoma), which is an extremely rare event. However, an in-depth molecular study carried out on both lymph node biopsy and BM samples with sorted populations of T and myeloid cells revealed the presence of the *SET::NUP214* fusion gene, *CBL* pathogenic mutation, and common *TCR* clonal repertoire in all analyzed samples, thus pointing to a common origin of tumor cells. Therefore, our data reveal significant plasticity of *SET::NUP214*-positive hematological neoplasms, with the ability of BM malignant progenitors to diverge and differentiate in different cellular microenvironments. Our observation emphasizes a leading role for molecular biology research in this unusual presentation of the disease.

## Figures and Tables

**Figure 1 ijms-24-14451-f001:**
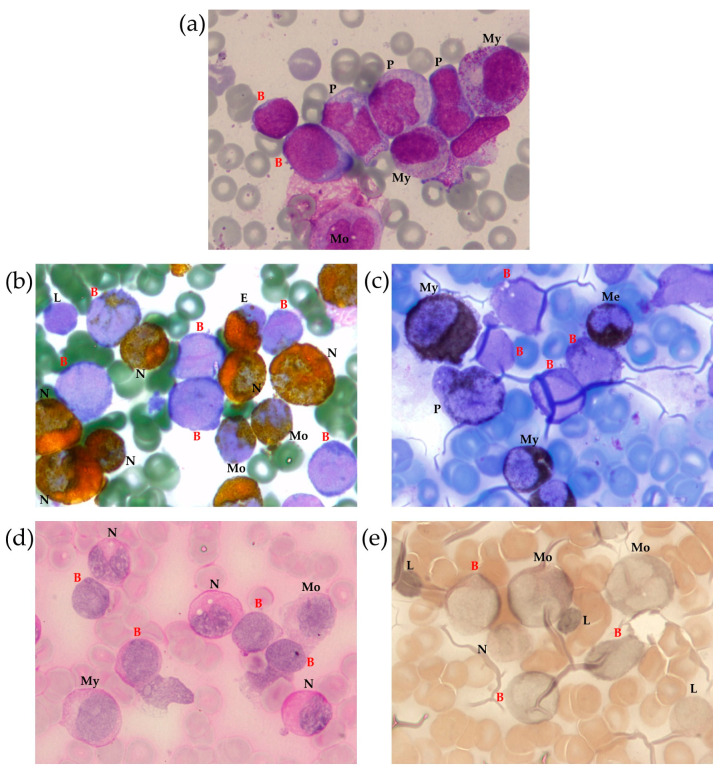
Bone marrow (BM) morphological examination with Romanowsky–Giemsa staining (**a**) and cytochemical examination for MPO (**b**), lipids (Sudan black staining) (**c**), glycogen (PAS reaction) (**d**) and non-specific esterase (**e**). Identified cells are marked as follows: B—blast, E—eosinophil, L—lymphocyte, Me—metamyelocyte, Mo—monocyte, My—myelocyte, N—neutrophil, P—promonocyte, 100× magnification.

**Figure 2 ijms-24-14451-f002:**
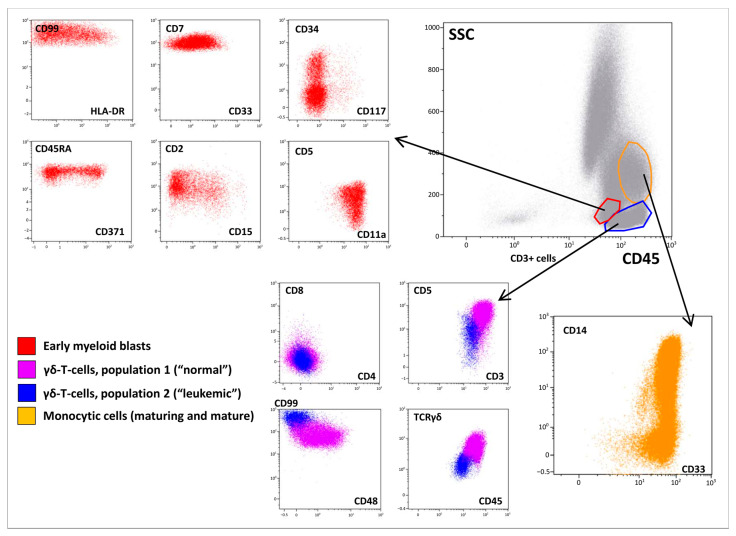
Immunophenotypic description of the bone marrow (BM). A BM overview is presented in the “gray” panel; a brief description of early myeloid blasts is presented in the “red” panel; two populations of γδ-T lymphocytes are shown in blue and violet, while maturation of monocytic cells is shown in orange.

**Figure 3 ijms-24-14451-f003:**
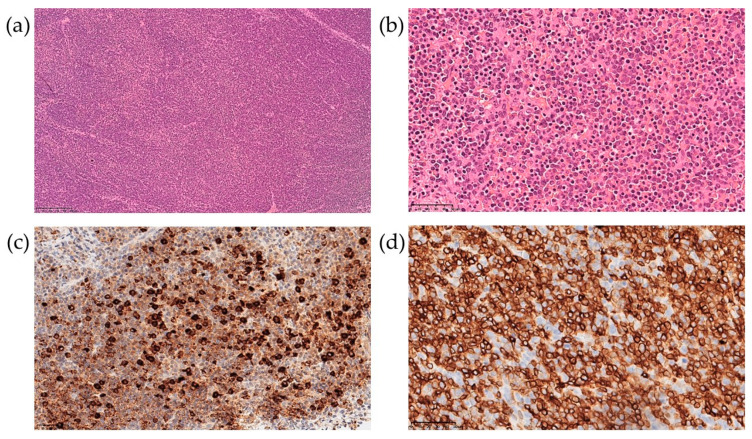
Lymph node biopsy examination with hematoxylin-eosin staining at 100× (**a**) and 400× (**b**) magnification, and immunohistochemical examination at 400× for MPO (**c**) and CD3 (**d**).

**Figure 4 ijms-24-14451-f004:**
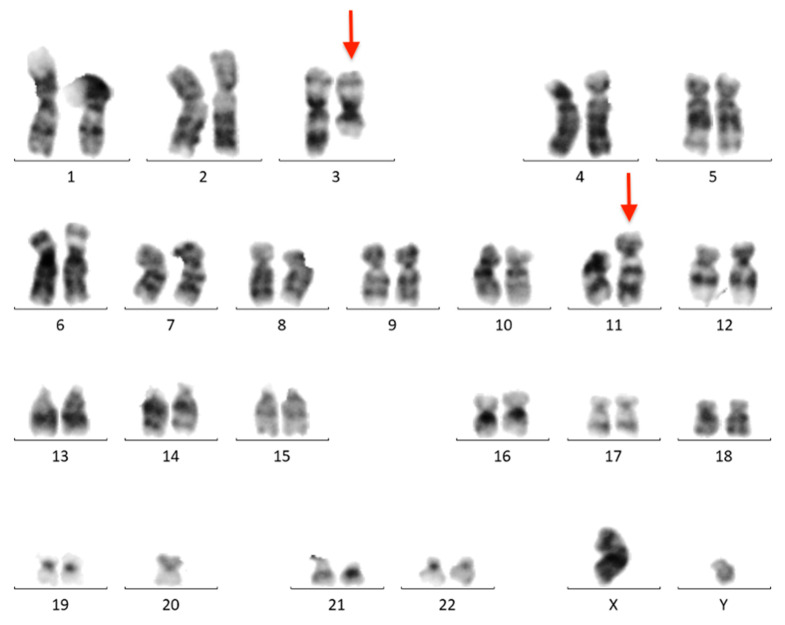
Conventional G-banded karyotype analysis showing the t(3;11)(q21;q23) translocation in whole BM. Rearranged chromosomes are marked with arrows.

**Figure 5 ijms-24-14451-f005:**
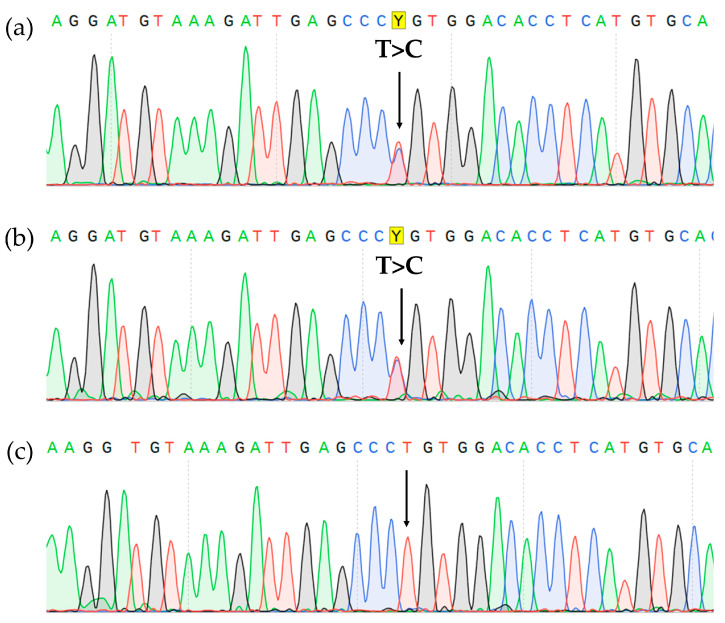
*CBL* gene, exon 8 Sanger sequencing showing c.1186T>C, p.Cys396Arg missense variant (highlighted in yellow) in the BM (**a**), lymph node (**b**), and it’s absence in the nails (**c**).

**Figure 6 ijms-24-14451-f006:**
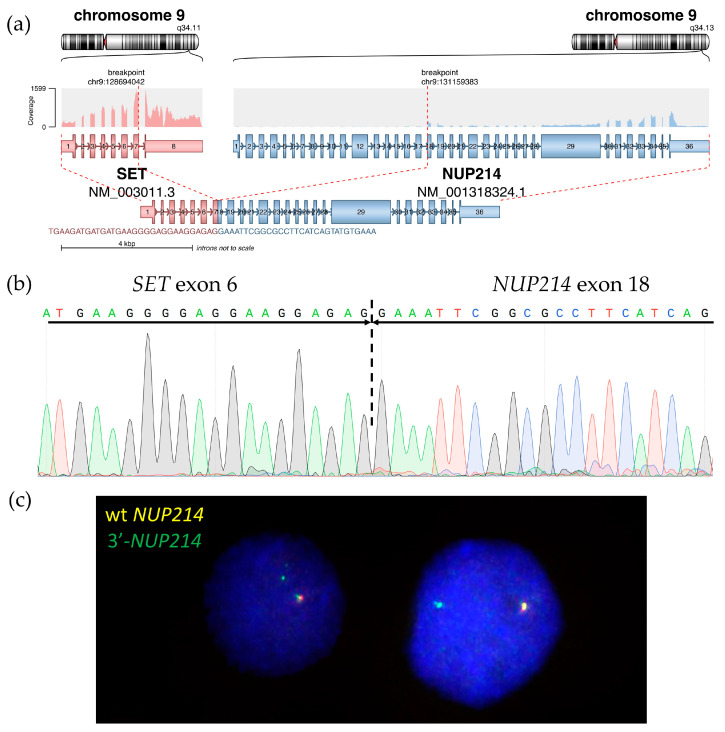
Molecular findings on *SET::NUP214* fusion in the BM: structure of *SET::NUP214* fusion transcript in RNAseq data as depicted using the Arriba algorithm (**a**) and Sanger validation (*SET::NUP214* exon 6—exon 18 junction is marked) (**b**); *NUP214* gene rearrangement with 5′-partial deletion confirmed using FISH with *NUP214* breakapart probe (CytoCell), 63× magnification, wild-type *NUP214* is yellow, rearranged *NUP214* 3’-portion is green (**c**).

**Figure 7 ijms-24-14451-f007:**
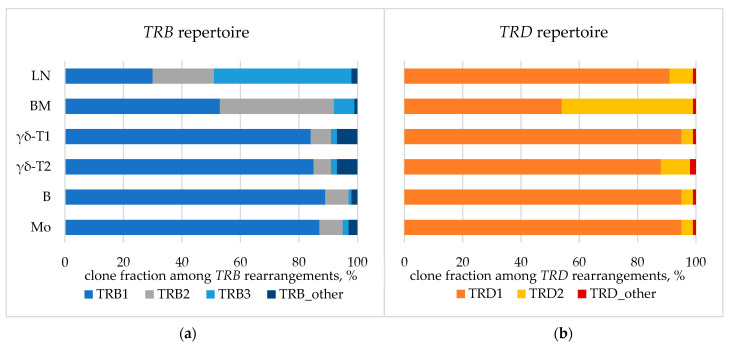
*TCR* repertoire: fraction of *TRB* (**a**) and *TRD* (**b**) clonal rearrangements in lymph node. BM—bone marrow aspirate, LN—lymph node biopsy. BM populations are as follows: B—early myeloid blasts (CD45^+^CD33^+^CD14^−^), Mo—maturing and mature monocytes (CD45^+^CD33^+^CD14^+^), γδ-T1—immature γδ-T lymphocytes considered leukemic (CD45^dim^CD3^dim^), γδ-T2—more mature γδ-T lymphocytes considered normal (CD45^high^CD3^high^).

**Figure 8 ijms-24-14451-f008:**
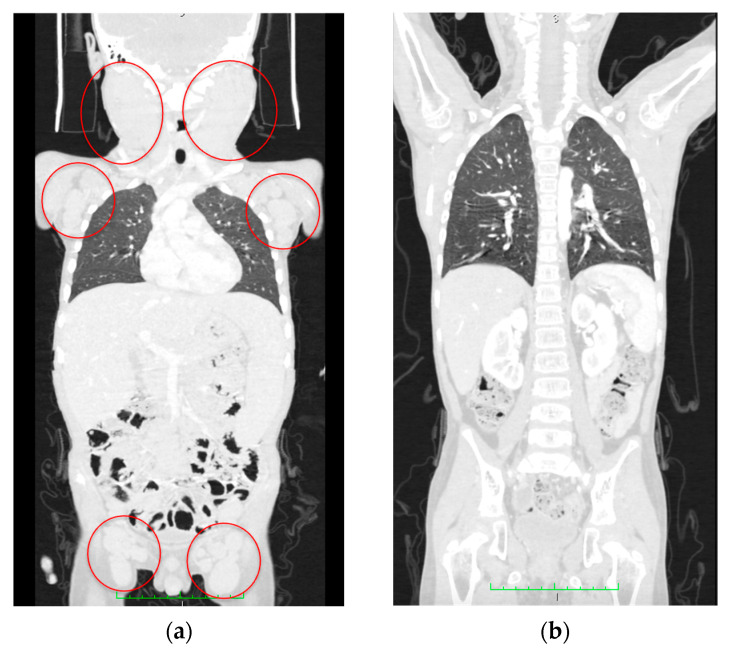
CT scan data. (**a**) Initial CT scan on admission with massive lymphoproliferative syndrome. Red circles indicate lymph node conglomerates. (**b**) Computed tomography at 180 days after hematopoietic stem cell transplantation; no lymphadenopathy and recession of the lymphoproliferative syndrome are seen.

**Figure 9 ijms-24-14451-f009:**
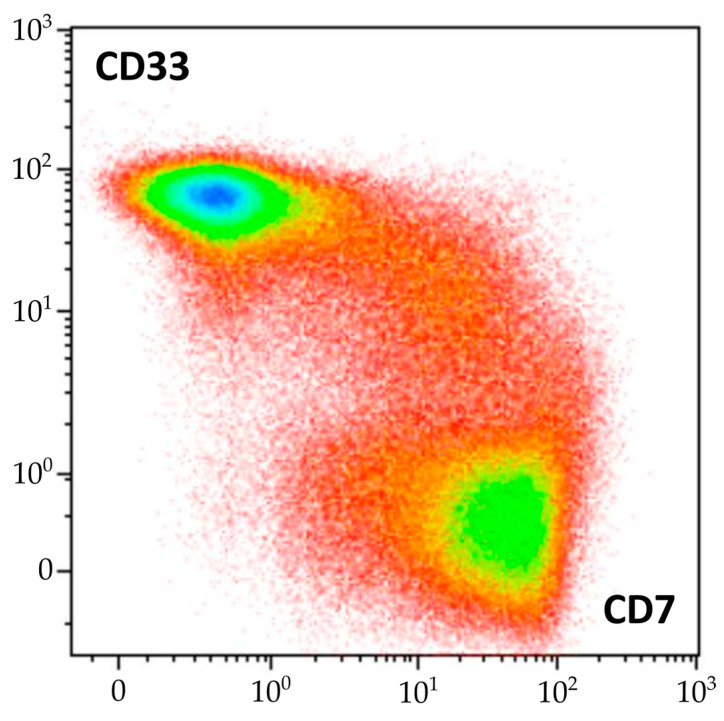
Continuity of the flow cytometric pattern of myeloid and T-lineage cells in BM of described patient, density plot of CD7 and CD33 expression (represented by color scale from red to blue). Mature neutrophils, erythroid precursors, NK cells, and B lymphocytes are excluded.

**Table 1 ijms-24-14451-t001:** Patient’s diagnostic examinations supportive of concomitant hematological neoplasm.

Examination	Acute Myelomonocytic Leukemia	PTCL
BM aspirate morphology and cytochemistry	A population of anaplastic blast cells positive for MPO and Sudan Black with a weakly diffuse PAS and negative for nonspecific esterase, 23% of BM NCs	-
BM aspirate immunophenotyping	1. A population of early myeloid blasts with CD45^dim^ expression, bright CD7, heterogeneous CD33, CD34, and CD117 positivity, and expression of CD2, CD5, CD99, HLA-DR, and CD15, 2.5% of BM NCs; 2. Monocytic cells at various stages of differentiation, with partial CD7^dim^ expression in the more immature population, 34% of BM NCs	A population of γδ-T cells with low CD7, CD2, CD5, CD3, and TCRγδ expression, CD45^dim^ and CD99^high^ expression, CD4/8-negative and CD48-negative, 3.5% of BM NCs
BM trephine biopsy	MPO-positive promyelocytic and myeloblastic components located predominantly in the centers of intertrabecular spaces	Lymphoid clusters positive for CD3
Lymph node biopsy	MPO-positive component	CD3-positive component

## Data Availability

The data presented in this study are available on request from the corresponding author. The data are not publicly available due to privacy restrictions.

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
