# Peer review of "Unusual Presentation of SET::NUP214-Associated Concomitant Hematological Neoplasm in a Child—Diagnostic and Treatment Struggle"

_ijms, 2023, doi:10.3390/ijms241914451_

Round 1

Reviewer 1 Report

This is an interesting case report of a child who presents with AML and concomitant peripheral T cell lymphoma. 

Comments

1. What criteria were used for diagnosis? WHO? If so please clarify

2. Please include an image of the lymph node.

3. Please clarify what was analyzed. If bone marrow aspirate first then please state this then trephine bx following this. As is it seems to go aspirate to lymph node to bone marrow bx. This is hard to follow. 

4. A table showing the two diagnosis would be helpful to better clarify how the diagnosis was made. 

5. Was single cell analysis done? If not then the persistent molecular findings could be a result of AMA contaminant. Also would be useful to show the negative control (ie skin) results. 

English is fine. Some minor grammatical changes throughout. 

Author Response

1. What criteria were used for diagnosis? WHO? If so please clarify

Diagnosis was clarified.

2. Please include an image of the lymph node.

The lymph node image was included as requested.

3. Please clarify what was analyzed. If bone marrow aspirate first then please state this then trephine bx following this. As is it seems to go aspirate to lymph node to bone marrow bx. This is hard to follow.

Examination information was restructured accordingly.

4. A table showing the two diagnosis would be helpful to better clarify how the diagnosis was made.

Table 1 was added to Case description subsection. 

5. Was single cell analysis done? If not then the persistent molecular findings could be a result of AMA contaminant. Also would be useful to show the negative control (ie skin) results. 

Single cell analysis was only performed for top TCR rearrangements in flow-sorted single nuclei from whole bone marrow aspirate to seek for co-existance of clonal rearrangements.
We used DNA from the nails as a negative control, and it was negative for CBL mutation (Figure 4 was modified accordingly) and SET::NUP214 fusion gene (Figure 1A was added).

Reviewer 2 Report

In the paper “Unusual presentation of SET::NUP214-associated concomitant malignant neoplasm in a child – diagnostic and treatment struggle”, by Yaroslav Menchits and colleagues, authors describe a single case with concomitant acute myeloid leukemia and peripheral T-cell lymphoma. Authors performed comprehensive morphological, immunophenotypic and genetic characterization of the tumors and demonstrate that the two tumors likely have clonal origin. Overall, this case report is well written and interesting for broad readership, including clinicians and basic scientists.

 Major remarks:

1. Discussion is quite long and often not related to the topic of the paper. I would suggest shortening it.

2. Is it possible to perform whole genome sequencing on these tumors in order to infer point of clonal diversification through a spectrum of shared VS unique genomic alterations?

3. Finding of CBL mutation in both tumors is quite interesting and may indeed indicate common (pre)malignant ancestor. Can authors exclude the possibility of germline mosaicism? Is it possible to check for presence of CBL mutation in other mesodermal tissues, old blood samples or Guthrie blood cards taken at birth for metabolic screening?

 Minor remarks:

1. All abbreviations should be explained when mentioned for the first time, e.g., CBC in lines 67-68.

2. What is the difference between figures 1a and 1b? Authors should give an explanation what is shown in both of these figures rather than just referring to figure 1 in the text.

The quality of English language is good. Minor spell check might be required. 

Author Response

1. Discussion is quite long and often not related to the topic of the paper. I would suggest shortening it

DEK-NUP214 and NUP214-ABL1 subsections of the Discussion were removed as irrelevant for the main topic.

2. Is it possible to perform whole genome sequencing on these tumors in order to infer point of clonal diversification through a spectrum of shared VS unique genomic alterations

This is a valuable suggestion. However, this is not possible since lymph node biopsy material is not available.

3. Finding of CBL mutation in both tumors is quite interesting and may indeed indicate common (pre)malignant ancestor. Can authors exclude the possibility of germline mosaicism? Is it possible to check for presence of CBL mutation in other mesodermal tissues, old blood samples or Guthrie blood cards taken at birth for metabolic screening?

The germline mosaicism was excluded after nail DNA examination. Figure depicting Sanger results was modified accordingly.

1. All abbreviations should be explained when mentioned for the first time, e.g., CBC in lines 67-68.

CBC was changed to blood test for the sake of readability.

2. What is the difference between figures 1a and 1b? Authors should give an explanation what is shown in both of these figures rather than just referring to figure 1 in the text.

Indeed there was no difference between figures 1a and 1b. Figure 1 was changed accordingly. For better description cytochemical reactions were added.

Reviewer 3 Report

In this manuscript, the authors present a rare case of concomitant presentation of acute leukemia of myeloid origin and peripheral T cell lymphoma. The most intriguing feature of this case is that the authors were able to prove the common origin of these two neoplasms that both presented the rare SET:NUP124 translocation. The discussion part is also of note, where the authors nicely review the literature on translocations involving NUP214, as well as the pathophysiological mechanisms implicated in the progression to hematological neoplasia. Some minor issues that could be improved are described below.

Comment 1

The authors should consider rephrasing the term “malignant neoplasm” in the title, so that the hematologic nature of the neoplasm is obvious to the reader (possible alternatives: hematologic neoplasm, hematological malignancy)

Comment 2

In figure 1 it is advisable to add arrows indicating the different cells described in the legend, that readers who are not hematologists can also discriminate them.

Comment 3

Line 86-87 “several cell populations did not appear to be definitely normal though lacked pronounced immunophenotypic aberrations”. Does this refer to the monocytic and T cell populations described in the following part of the text? Please define. If it applies to other cell populations it would be preferable to briefly describe them.

Comment 4

In figure 2 for the SSC-CD45 plot it would be preferable to draw the gates for blasts, lymphocytes and monocytes.

Comment 5

Please consider providing also the myeloid leukemia classification based on the latest WHO edition (only FAB classification is provided).

Author Response

1. The authors should consider rephrasing the term “malignant neoplasm” in the title, so that the hematologic nature of the neoplasm is obvious to the reader (possible alternatives: hematologic neoplasm, hematological malignancy).

This is a valuable comment certainly clarifying our main point. We thank the reviewer for this comment and the title was changed accordingly.

2. In figure 1 it is advisable to add arrows indicating the different cells described in the legend, that readers who are not hematologists can also discriminate them.

Figure 1 was changed accordingly. For better description cytochemical reactions were added. Identified cells were marked.

3. Line 86-87 “several cell populations did not appear to be definitely normal though lacked pronounced immunophenotypic aberrations”. Does this refer to the monocytic and T cell populations described in the following part of the text? Please define. If it applies to other cell populations it would be preferable to briefly describe them.

Yes, this sentence referres to monocytic and T-lineage compartments. We have modified it accordingly.

4. In figure 2 for the SSC-CD45 plot it would be preferable to draw the gates for blasts, lymphocytes and monocytes.

Drawing the CD45/SSC dot plot, we aimed to show the whole structure of the cell distribution in BM with the links between different cell compartments. Therefore, we prefer not to paint different cell types with different colors on this plot. In order to make general gating clearer, we have added the gates drawn with thin lines of the respected colors, although we needed to add one more clarification to the lymphoid compartment gating because only T-lymphocytes were the subject of analysis.

5. Please consider providing also the myeloid leukemia classification based on the latest WHO edition (only FAB classification is provided).

Diagnosis was clarified.